# Towards a “New Mothering” Practice? The Life Experiences of Mothers Raising a Child with Autism in Urban Ethiopia

**DOI:** 10.3390/ijerph20075333

**Published:** 2023-03-30

**Authors:** Rahel Fentahun Asmare, Fasil Nigussie Taye, Messay Gebremariam Kotecho, Faye Mishna, Cheryl Regehr

**Affiliations:** 1School of Social Work, Addis Ababa University, Addis Ababa P.O. Box 1176, Ethiopia; rachelf833@gmail.com (R.F.A.); fasilfnt@gmail.com (F.N.T.); mesghe@gmail.com (M.G.K.); 2Department of Social Work and community development, University of Johannesburg, Johannesburg P.O. Box 526, South Africa; 3Factor-Inwentash Faculty of Social Work, University of Toronto, Toronto, ON M5S, Canada; f.mishna@utoronto.ca

**Keywords:** autism spectrum disorder, stress, mothering, new motherhood, acceptance, resilience

## Abstract

Autism spectrum disorder (ASD) is a complex neurological and developmental disorder that has seen an increase in prevalence over the past two decades, particularly in low and middle-income countries. The purpose of the current paper is to examine the experiences of mothers in Ethiopia raising a child with ASD through employing a qualitative research design involving semi-structured interviews with twenty mothers. The experiences of mothers in this study fell into three thematic areas: (1) grieving and experiencing other emotions arising from the diagnosis of their child; (2) developing, understanding and defining autism; and (3) accepting the diagnosis and developing coping strategies for raising their child. The findings revealed that raising a child with autism introduced a new lifelong experience to mothers’ everyday lives, profoundly changing their parenting role and transforming their view of mothering. Recognition of the experience of “new mothering” and mothers’ meaning-making process, stress, coping mechanisms and resilience is critical to informing policies, programs, counseling and other therapeutic efforts to assist children with autism and their families for social workers in Ethiopia and those working with the Ethiopian diaspora in other regions of the world.

## 1. Introduction

Autism spectrum disorder (ASD) is a complex neurological and developmental disorder that involves persistent developmental challenges causing mild to significant levels of impairment in speech, non-verbal communication, social interaction and behavior [1,2,3]. While there has been an increase in the prevalence of ASD globally over the last two decades [4], the vast preponderance of research has been conducted in Western high-income countries, essentially ignoring the experiences of 80% of the world’s population [5].

The global prevalence of ASD has been estimated to be between 1–2% of the population [6,7], and evidence shows heterogeneity of prevalence estimates of ASD across different geographic areas, due to different methodologies that are being applied in case detection [8] and sources of data used to estimate prevalence [4]. Yet, the rates of ASD prevalence and its co-morbidities in low- and middle-income countries are largely unknown due to limited resources, scarce knowledge of ASD, and weak healthcare systems [9,10].

Parents whose child is diagnosed with ASD face multifaceted difficulties, including: economic burden related to income loss and high costs of rehabilitation [11,12]; less time for leisure and entertainment [13]; altered family dynamics and frequent parental conflicts [14]; and problematic behaviors and distressing emotions [15]. When compared to parents of children with other developmental disabilities, parents of children with ASD report higher levels of stress [16], a burden largely borne by mothers [3,17], and more mental health problems including depression and anxiety [18,19]. While they may experience higher levels of stress, support from family and friends or other support such as professional guidance [5,20,21], as well as looking for positives in the experience [22], serve as buffers. In overcoming stress, mothers employ various coping strategies including active avoidance coping, positive coping, problem-focused coping and religious/denial coping [23].

Parental experiences of raising a child with ASD vary according to contexts including location, family size, marital and occupational status, social support and socioeconomic conditions [9,17,24]. The bulk of research on the parental experiences of raising a child with ASD, however, has been conducted in Western and high-income countries, with few studies in low- and middle-income countries [5,25]. Little is known in low-and-middle income countries about the parental experiences of raising a child with ASD [4,26,27]. The limited research that exists reports that parents have limited awareness of the diagnosis and a lack of access to information and education on ASD [28,29]. Not knowing how to help their child with ASD contributes to their distress, strain on the marital relationship and poor parenting.

In resource-poor countries, raising a child with ASD and other types of mental disorder is more challenging and complex due to a lack of diagnostic and educational services, and due to limited parental awareness. Recent studies have described the experiences of parents of children with ASD in Nepal, where parents and healthcare providers have little awareness of the illness [30], and in India, where mental illness and disability have historically been kept hidden, largely as a result of stigma [31,32]. In Egypt, to care for a child with ASD was found to be daunting and overwhelming to parents due to a lack of awareness, as well as due to minimal or even absent services for children with ASD [33]. In Jordan, the most common challenges that parents faced in caring for their children with ASD were a lack of public awareness about ASD, financial burdens on parents, and limited specialized knowledge among healthcare providers about ASD [34].

Unravelling mothers’ experiences of caring for a child with ASD is crucial to facilitate effective interventions. This is particularly important as research that focuses on maternal experiences of caring for children with ASD is rarely addressed in low- and middle-income countries [33]. In the current study, using a qualitative methodological approach, we investigated the experiences of mothers whose child was diagnosed with ASD and who identified aspects of raising a child with this diagnosis. The aim is to contribute to the growing research that documents parental experiences and coping mechanisms in raising a child with ASD in the global south, specifically shedding light on the experiences of Ethiopian mothers. Studying the experiences of mothers is particularly important as they have a profound role in raising the child. The study was guided by the following research questions: a) What are the experiences and reactions of mothers during and after the time of their child’s diagnosis with ASD in urban Ethiopia? B) How did mothers of children with ASD develop coping strategies to strengthen their resilience against stress in urban Ethiopia?

### 1.1. The Ethiopian Context: Parental Experiences in Raising a Child with ASD in Ethiopia

Considering the relevance of contributing to the limited but growing research that documents parental experiences of raising a child with ASD in developing countries, this study was conducted in Ethiopia, a country located in the horn of Africa. The second most populous country in Africa, Ethiopia has a total population of over 120 million, with children representing more than half of the population. As there are no official statistics, the prevalence of ASD among Ethiopian children is unknown. Ethiopia is among the least developed countries in the world with extremely limited mental healthcare facilities available. As most of the Ethiopian population (over 80%) lives in rural settings, diagnostic and treatment services to children with ASD are largely non-existent [35,36]. The very few available educational and diagnostic services for children with ASD are concentrated in the country’s capital, Addis Ababa [27].

To overcome the problems of low coverage of mental health services in rural Ethiopia, in 2003, the Ministry of Health implemented a community-based mental health services model as part of the health extension program [35]. Over 38,000 health extension workers (HEWs) who completed a one-year basic health training course have been deployed throughout rural Ethiopia providing primary healthcare services with health promotion and prevention packages. Through this strategy, mental healthcare services are provided by non-specialist HEWs [37]. Although not trained to diagnose and treat mental health problems, the HEW’s have played a vital role in ensuring decentralized care provision, which has been found to be promising as a support for parents who have children with ASD [36].

Social workers are increasingly recognized to be working in the Ethiopian healthcare system and hospital settings. Social work practice in Ethiopia is very different, however, from the established practices in resource-rich countries in the global north. Informed by the bio–psycho–social–spiritual model, social work practices in many developed countries (including the United States) center the mental health needs of individuals, families, groups, and communities [38]. In contrast, social workers in Ethiopia do not commonly deal with mental health issues. Rather, much of the focus of social work practice in Ethiopia is influenced by policy priorities and the government’s political ideology. Informed by poverty reduction policies and strategies, the areas of practice for many of the Ethiopian social workers have recently included social protection and poverty reduction. Many social workers practice social welfare in NGOs, and to a lesser extent in hospital settings. Consequently, the lack of professional healthcare workers, including mental health social work practitioners, has affected service provision to children with ASD and their parents.

After interviewing service providers in Ethiopia, Tekola and colleagues (2020) concluded that services for children with ASD are extremely limited, and that stigma and lack of awareness are impediments to overcome [27]. The bulk of care consequently falls to families, and in particular, mothers. Despite occupying a lower socioeconomic status and being excluded from decision-making roles in families, communities, and organizations, women in Ethiopia often play an instrumental role in family and community affairs, including raising children with ASD. An emerging research literature reports the challenges mothers face in raising a child with ASD, including spousal abandonment [39], quitting their jobs to care for the child [40] and social stigma [36]. Many mothers are reluctant to leave their children with relatives or neighbors because their child’s behaviors are typically misunderstood and considered a result of either poor parenting or punishment for sin. Talking about their children can be difficult, as mothers may feel guilt and shame. They also fear social stigma, which greatly affects the lives of children with ASD. Children with ASD are among the most disadvantaged social groups because physical and mental impairments are frequently viewed by Ethiopian society as related to such factors as a curse or bad omen [41]. Thousands of children with ASD are confined to their homes with little access to education and rehabilitation [27]. Educating a child with ASD is not a priority in a country with pervasive social stigma. Furthermore, the problems are compounded by the lack of awareness of many professionals in the medical, educational and vocational fields regarding how to effectively work with individuals with ASD [29,36].

There are, however, cultural resources that can be utilized to increase awareness of ASD and reduce stigmatization. Weldeab and Opdal (2007) noted that some traditional practices in Ethiopia can be used as sources for change [41]. For instance, social gatherings (e.g., coffee ceremonies) can be a space for dialogue through which societal awareness can be increased. Indigenous social institutions and self-help associations (e.g., idir, equab, senbet, mahber), established by community members to facilitate social insurance and economic support, can be spaces in which to increase awareness and challenge the negative stereotypes of ASD.

### 1.2. Theoretical Lens: Mothering

In the current study, the theoretical lens is based on the scholarly work of Barlow and Chapin (2010), in which mothering is understood as an emergent process that relates to the practices in which women participate by virtue of ‘being a mother’ [42]. Although the practice of mothering is key in the construction of gender roles, social status, kinship and identity, it is shaped by specific social contexts that vary based on material and cultural resources and constraints [43]. Mothering thus has multiple and often shifting meanings according to time and place [44]. While theoretical discussions on mothering abound, few studies analyze the challenges of developmental disorders on parents using the concept of mothering. The theoretical lens of mothering is therefore operationalized in the current study as a means to understand the cognitive, psychosocial and emotional processes of change that mothers experience when their child is diagnosed with ASD [45].

## 2. Methods

Using a qualitative research design [46], data for the current study were collected by the first and second authors, using semi-structured interviews. The interviews aimed at developing insight into the understanding and experiences of mothers in urban Ethiopia who have children with ASD. Interviews of 30–40 min in length were conducted either through face-to-face interactions with the mothers, or via telephone following an interview guide. Interviews were conducted in the Amharic language to allow for the broadest participation and to capture participant views in the language of their experiences. All interviews were tape recorded and transcribed. Originally transcribed in Amharic, the transcripts were then translated into English. The tapes and transcripts were stored in a safe place that was only accessible to the researchers.

Ethical approval was obtained from the Ethics Committee of Addis Ababa University’s College of Social Sciences. Following a verbal description of the study, all participants provided oral consent to take part. This consent was recorded and transcribed. To conceal the participants’ identities, pseudonyms are used in this manuscript.

To recruit the participants, we approached two civil society organizations that provide support to children with ASD and their parents. The support provided by these agencies includes treating children, training parents in raising their children and providing day care services. Most support is free with some financial contributions from parents based on capacity and willingness to pay. Purposive sampling was utilized in selecting participants. The selection criteria included (a) mothers who were raising a child with ASD; (b) mothers who provided care to their child with ASD for more than two years; (c) mothers who were willing to take part in the research; and (d) mothers who received services from Bright and Joy Autism centers.

### 2.1. Participants

A total of 20 mothers participated in the study. The average age of the mothers was 39.8 with an age range of 30 to 57. Eighteen participants were married and two reported that they were divorced. Five mothers had some or no secondary school education, seven reported that their highest level of education was a secondary school diploma, five had undergraduate degrees, one had a graduate degree and two had professional degrees (in teaching and medicine). Eleven participants reported their occupation as homemaker, six were professionals (nurse, teacher, physician, accountant, program officer), two worked in service jobs (cleaner, cook) and one was a business owner. The average age of children with ASD at the time of the study was 11.8 years (range 4–17); the average age at diagnosis was 4.5 (range under 2 to 9 years). Eighteen of the children were male and two were female.

### 2.2. Data Analysis

Data were analyzed using thematic analysis to understand how mothers of children with ASD make sense of their personal and social worlds. The transcripts were read repeatedly to familiarize the researchers with the data. Next, significant statements were identified using the selective highlighting approach [47], which entailed choosing and highlighting key words, phrases and sentences that significantly stood out and illustrated common experiences. Significant statements were coded, and the coded data were analyzed to identify major themes. Through this thematic analysis, we extracted three themes that best exemplified mothers’ experiences of raising a child with ASD.

### 2.3. Trustworthiness

Trustworthiness in qualitative research has traditionally focused on verisimilitude or the appearance of truth [48], that is, achieving a sense of resonance or congruence with the audience that may have experienced similar situations [49]. An additional suggested criterion is the utility of the narrative in terms of assisting with comprehending an experience and enhancing a group’s future problem solving [50]. In the current study, the authors engaged in extensive re-reading of the interview transcripts and checked for the accuracy of translation of the interview transcripts from Amharic to English. As the data analysis process unfolded, emerging themes were discussed with expert peers and compared with the existing literature. The authors also verified the credibility of findings by having discussions with selected mothers and double-checking that the identified themes properly captured their experiences. In this respect, our forms of trustworthiness included prolonged engagement, triangulation, peer validation and member checking [50,51].

## 3. Results

In this section, the extracted themes and subthemes of the study are presented, including mothers’ emotional reactions during the diagnosis of their child with ASD; their cognitive responses after diagnosis, which involves (re)defining ASD and acceptance of their child’s mental health status; and, finally how they have developed new mothering practices through reframing their relationships with their child who has ASD and identified positive learning experiences. Throughout this section, pseudonyms are used to add a human face to the experiences of the mothers while protecting their identity.

### 3.1. Shock and Grief: Emotional Reactions to Diagnosis

Prior to receiving their child’s diagnosis of ASD, many participants described having sought appointments at various medical centers, seeing numerous doctors/pediatricians and other healthcare professionals to understand the nature of the challenges they witnessed in their child, and seeking assistance to address these challenges. Our findings show that during the investigative and diagnostic process, mothers experienced an emotional rollercoaster including sadness and despair. After receiving a diagnosis of ASD for their child, the participants described a grief-like process, whereby they felt they had lost or were losing their child. They reported myriad responses, including hopelessness and shock, and described asking, “why me?” For instance, Ayelu explained that she cried, Dagmawit said she felt shocked and numb, and Kebebush felt confused. Many participants said they asked God why their child was selected to have ASD. Having described her child’s behaviors, Enatenesh expressed, “When I knew it was ASD, there was no way out! I was so distressed, depressed and felt despair. I asked God Lemin enen” (literally: God, why me?).

The participants described feeling overwhelmed by the prospect of this diagnosis with little information, medical support or referrals. Lulit revealed feeling “devastated and helpless” upon hearing her child had ASD. She explained that she was 14 and unwed, about which her family was unhappy. Having a child with ASD was overwhelming and an extra burden.

### 3.2. (Re)Defining ASD: Cognitive Understanding and Meaning-Making

Determining the causes of ASD emerged in the analysis of the interviews as a meaning-making process that shaped the participants’ maternal thinking and practice. In most instances, the mothers’ assumptions about the causes of ASD negatively affected their worldviews, including feeling that they were losing their child. The participants’ struggles to make meaning of their child’s diagnosis was an extended process in which they interpreted and acted upon experiences related to their child. For example, while Wolansa’s child was diagnosed with ASD in 2014, six years prior to the study, she exclaimed, “there are not enough words to define Autism.” She referred to the stigma as well as the financial and lifestyle burdens and depicted ASD as “a dreadful kind of disorder”, adding, “I feel like I am living in another new world.” Sable similarly depicted ASD as “a terrible disorder.” She explained, “due to Autism, my daughter knows very little about her surroundings. This cold truth makes me live in a dark state of mind”.

There is little awareness of ASD in Ethiopia [36]. Several mothers reported having limited information at the time of their child’s diagnosis with ASD, and none of the participants reported receiving counseling either during or after the diagnosis. There is an acute shortage of professionals in Ethiopia trained to provide services to children and their families who are affected by a developmental disorder [27]. The few available healthcare services are in urban areas and are expensive. Mothers therefore found it difficult to know how to access information or treatment and services. For example, although Kelemua had not heard the term autism before hearing her child’s diagnosis, she was not provided information or psychological support. Explaining that she was just prescribed medication to help her child sleep, Kelemua commented, “I really did not know what to do and where to go”. Another participant, Demeku, explained that even though she had been working as a nurse for several years, she was unable to comprehend her situation. She worried about what she was going to do and about what her son “would become.” She explained that she “felt like I was left in an isolated island. No one lifted me; no satisfactory information, it was also very difficult to find an Autism center that provides support in our country”.

The meaning-making process entailed participants searching for the causes of ASD. As presented in Figure 1, the participants reported a range of possible causes that resulted in their child developing ASD.

The mothers reported God’s intention as the most frequent cause (*n* = 5) for having exposed their child to ASD. Accordingly, several mothers turned to religious coping so their child could be healed by supernatural powers. Mastewal explained that her son was born healthy with no problems during delivery; his development was normal until he was two years. She explained that an act of God was needed to change her child’s mental health condition. Likewise, Kelemua reported that she did not do anything wrong during her pregnancy and that the fetus was healthy, noting that her son was okay until the age of three years. She similarly believed her son developed ASD as a punishment from God because of a sin by her family.

Two mothers were teenagers (14 and 16) when they gave birth, which they each thought could be the cause of their child’s ASD. As noted above, Lulit was 14 years when she gave birth. As her father refused to see her anymore, she left her parents’ home and lived on the street during the first seven months of her pregnancy. While her mother tried to help her when she could, Lulit could not care for herself during the pregnancy. She believes that if she had regular medical checkups and family support, her daughter would not have been diagnosed with ASD.

Other causes of ASD were attributed to the actual birth. Premature birth was one factor believed to have caused their child’s ASD. According to one participant (Silanchi), oxygen could not flow adequately to her baby’s brain due to respiratory distress and breathing difficulties. Another factor according to one participant (Serkadis) entailed the medical mistakes nurses made during delivery. Other mothers (*n* = 3) believed the cause of their child’s ASD was genetic predisposition, as evidenced for example by several children in the extended family also having a developmental disorder (Demeku). Less common reasons included the child‘s exposure to ASD due to contraceptive pills taken early in the pregnancy (Dagmawit), or to chemicals and drugs during the pregnancy because of their work (e.g., as a veterinarian; Enatenesh).

### 3.3. Acceptance, Coping and Moving Forward

After experiencing prolonged emotional reactions and (re)defining the causes of ASD in their own terms, several participants commented that they accepted their child’s status. They found acceptance particularly important in facilitating their emotional transition from despair to hope, boosting their resilience, developing coping strategies and adopting new mothering practices. They began considering their child’s condition as unique rather than pathological. Silanchi relayed that over time she realized that children with ASD are unique. She expanded that while her child could fix metals and open any locked door, he could not open his water bottle. She concluded that ASD was not a disorder but rather, “it is just uniqueness. The only thing the autistic children expected from us is to understand them. In my case, I just moved forward to train my boy and to let him be the best person in the world”. Meselech similarly talked about coming to accept her child’s ASD. She explained that while it took her a long time to align her lifestyle with her child’s needs, she began to change her mindset as well as her home setting: “I accepted it! I decided to change our lifestyle. I shifted everything based on my child’s needs. I have to learn to let that go, I learn to appreciate what he is doing”.

### 3.4. New Mothering: Reframing Relationships with Their Child with ASD

Many of the informant mothers (*n* = 16) revealed that they had begun to look forward after their child was diagnosed with ASD and that they learned to employ various strategies to train their children to meet developmental milestones. Enatenesh relayed having to start from scratch to understand her child’s emotional needs as she was juggling responsibilities as a researcher, a wife and a mother of other children. She stated that the experience of raising a child with ASD is ‘addis enatnet’ (literally meaning new motherhood). The participants believed this new mothering process changed their personality and worldview. Adaptation to new mothering required reframing goals, adjusting behaviors and developing new life skills. At the same time, raising their other children influenced their understanding of motherhood.

Mothering is a highly emotional and a deeply psychological process in which women develop close relationships with their children. A shift in mothering practice has emotional, economic, social and psychological components. What emerged in the analysis was a shift in the ways the participants acted and experienced being a mother, after their child was diagnosed with ASD. A shift in the participants’ mothering practice began as a change in their everyday lives. About half of the informant mothers (*n* = 10) left their jobs and became fulltime mothers and homemakers after their child’s diagnosis with ASD. Representative of these participants, Emnet quit her job to care for her child following the diagnosis of ASD twelve years earlier. She has a college degree (BA in business education) from Addis Ababa University and had a well-paying job in a private company. With Emnet’s spouse currently the family’s only source of income, this shift in mothering practice had economic effects as well as effects in other ways and areas.

Some participants initially found it difficult to form an attachment with their child, as they did not know what parenting style could work. These women described focusing on their child and putting the child’s needs first. For example, one participant (Wolansa) began to “recognize what set him off, what calmed him down.” She described her child as “still fairly non-verbal and he likes to be alone; it can sometimes be difficult to understand what his needs are. Yet we have our own ways of connecting and having fun with each other”. Wolansa was dedicated to helping her son learn life skills, for example brushing his teeth, which meant telling and showing him. She noted, “teaching basic life skills takes and needs a lot of energy and time, as I experienced it. It also makes me feel closer to him”. Another participant (Demeku) similarly found it was necessary to speak clearly to her child and to perform some important activities with him so that he could understand her and learn necessary life skills, for example, following a toilet routine.

For several participants, particularly those who were working, feeling emotionally connected with their child who had ASD was time consuming and required considerable effort. For Serkadis, “new motherhood” increased her burdens. Her daily routine began by cooking breakfast for her family followed by taking her child to the autism center. She later took him home after finishing chores and took care of him while performing other household activities. She found this routine “very depressing”, as there was no time to take care of herself. Another mother, Mastewal, reported that raising her child with ASD was a “nightmare” initially. Only after a long process of stress and adjustment did she slowly begin to reconstruct a new motherhood worldview and mothering practice. Silanchi likewise explained that with time she recognized that her child was “totally different”, which helped her to understand his reactions and to develop patience in recognizing his needs and wants. She said, “when you become a mom of an autistic child, you need to be a researcher in order to understand your child’s behavior and to figure out everything”.

### 3.5. Positive Learning Experiences and Cognitive Reframing

Despite the many challenges, several participants described personal benefits that would not have been possible had they not had a child diagnosed with ASD. Meselech explained that raising her child taught her time management and budgeting, increased her patience and understanding, and helped her to appreciate the small things in life. According to the mothers, it was difficult to learn these things. Meselech explained, “it was not easy to learn these things though. I learned after lots of ups and downs, passing through anger, tears and anxiety. My lifestyle is now changed, and I know the priorities in my life”. Kelemua concluded, “the most important thing I have learned is patience, patience, patience!” She added that she also became more empathic towards other children with developmental disorders.

## 4. Discussion

The current study investigated the experiences of mothers in Ethiopia who are raising a child with ASD. The findings revealed that in response to their child’s diagnosis with ASD, mothers initially experienced sadness and despair, and that their extremely limited knowledge of ASD prior to the diagnosis contributed to them feeling overwhelmed. This finding is in line with studies examining the experiences of mothers of children with ASD in other regions of the world [5,17,24,52,53,54]. These studies found that parents’ limited awareness of ASD contributed to initial reactions of denial, sorrow, fear and shock, and that parents who have children diagnosed with ASD exhibit greater levels of mental health problems such as anxiety and depression [18,19], as well as suicidal ideation [55].

The mothers in the current study all reported having received very little psychological or mental health support or guidance from healthcare professionals [27]. This has been a barrier in seeking treatment and services, thereby adding to the mothers’ stress. This finding is consistent with other studies in the Sub-Saharan African context which show that parents are displeased by the limited information and services they receive from healthcare professionals during their child’s diagnosis with ASD [56].

Determining the cause of ASD emerged as an important component of the meaning-making process that shapes maternal thinking and practice. Associating the causes of ASD with supernatural power is common in Africa. Other studies [26,41] reported that parents attributed the cause of having a child with ASD to evil spirits, punishment from God and witchcraft. Similarly, some of the mothers who participated in the current study ascribed their child’s diagnosis to God’s intention.

Studies have reported that the causes of ASD can be traced to both genetic and environmental factors [26,57,58]. Genetic inheritance, labor complications, viral infection during pregnancy, problematic infant immunization, organic brain defects and head trauma are presented as causes. The mothers in the current study believed that the causes of having a child with ASD included genetic inheritance, complications during delivery, lack of self-care, premature birth, exposure to drugs and taking contraceptive pills during pregnancy. Two mothers believed the cause might be a lack of self-care, illness and complications during pregnancy. This finding coincides with Gardener and colleagues who reported that medical complications during pregnancy may increase the risk of having a child with ASD [59].

A key finding is that the diagnosis of a child with ASD profoundly shifted the participants’ mothering roles and worldviews. Several mothers stated that after learning their child’s diagnosis, they found themselves living in a new world. Similarly, Hoogsteen and Woodgate (2013) found that parents were in their own world for all aspects of daily living and challenges related to parenting a child with ASD [60]. All mothers in the current study explained that the diagnosis of their child led them to adopt a different lifestyle and to change the family dynamics, as well as their relationship with their child. Some even used the term addis enatnet (literally new motherhood) to describe the shift in the mothering role after the diagnosis of their child. In an emotional and relational sense, a shift in mothering practice is expressed through reframing maternal relations with the child who has ASD. The new motherhood experience enabled mothers to explore different ways of parenting and to learn life skills to help them manage multiple roles.

Consistent with previous research [60,61,62], the findings of the current study showed that despite the challenges in their parenting roles, several mothers felt they benefited in some ways, including developing essential life skills such as patience and redefining their priorities. They were also able to develop other important life skills such as industriousness, appreciating the little things in life and increasing their empathy towards others. Many informant mothers described their child’s positive contribution in this regard and to their life-long journey as parents, which is supported by previous research [62]. The positive learning experiences boosted the mothers’ resilience, helping them to cope. Specifically, many mothers developed patience, which greatly contributed to their ability to reframe their relationship with their child and thereby help them regain a sense of control over their life.

### 4.1. Limitations

The researchers recruited a relatively small number of mothers who had children with ASD who were enrolled in two non-governmental charity organizations located in two cities in Ethiopia. As a result, the study has a limited scope in terms of revealing the experiences of mothers who live in rural areas, and whose children receive support from other institutional settings. Additionally, a key limitation of this study is it cannot capture all the different types of ASDs, including all the core ASD features and associated symptoms that children with ASD display. ASD consists of diverse abilities and disabilities, and the results of this study primarily correlate with mothers’ emotional distress and coping mechanisms. This study could not cover all the dimensions of the communication levels between mothers and their child with ASD and the child’s functionality. In addition, as the informants were mothers, the study omits the fathers’ perspectives, thus providing only a partial story of parental experiences of raising children with ASD.

### 4.2. Implications for Mental Health Social Work Practice

The findings of this study align with the view that social work professionals need to continue engaging in a cross-fertilization of concepts and ideas that originate in different fields of study, including disability and gender studies. An interdisciplinary team is necessary to address the multi-dimensional challenges faced by mothers in raising a child with ASD in Ethiopia. As the mothers articulated, they experienced psychosocial challenges during and after the diagnosis. It is critical for social workers to collaborate with other professionals in hospital settings to provide appropriate mental health support and guidance about ASD, as well as strategies to handle child behavioral and mental health problems.

Our findings particularly inform professionals in Ethiopia working with families to understand the care burden of mothers in raising children with ASD. Social workers can play a meaningful role in helping families that have children with ASD through counseling support. Yet, the findings of this study indicate that supporting families in Ethiopia needs to go beyond counseling. Social workers may need to consider working with families to find alternative mechanisms in redistributing parental roles and easing the care burden that mothers carry, as well as find ways to ensure that mothers and their child with ASD are safeguarded from harm.

For professionals who work with the Ethiopian diaspora, the findings point to the importance of grounding their practice with a sound understanding of family dynamics and the influence of culture on the caring of children with ASD. As such, in supporting diaspora families and in mobilizing support, professionals may need to consider the meanings mothers attribute to their child’s challenges, how mothers carry the brunt of the burden, and how their everyday lives and their motherhood worldviews are altered after their child is diagnosed with ASD.

According to our findings, the mothers identified the centrality of strengths, capacities and potentials that enhance their resilience and ability to access and utilize support and resources from family members, hospitals, autism centers and religious organizations. It is critical for social workers to facilitate access to resources to strengthen mothers’ resilience and to advocate for such resources as income-generating activities, which can provide the mothers with the means of tackling their financial burdens. Importantly, social workers must focus on the rights of these children in accessing education and services, in order to improve their wellbeing.

Further research is required to increase social workers’ understanding and ability to work with mothers raising children with ASD in Ethiopia. Research examining support systems is vital to provide an understanding of the areas in which clients feel supported and those in which more support is needed [63]. Research on the experiences of these “new motherhood” practices, which uncover stressors, needs, difficulties and resilience, will inform social workers in meeting the needs of mothers raising a child with ASD.

Children with ASD and their mothers face many challenges in hospital settings. Despite the inclusion of mental health in the country’s national health policy, children with ASD and their mothers have not received sufficient support or treatment in healthcare settings. Further development and endorsement of mental health policy is thus necessary to ensure the responsibility and accountability of healthcare workers in hospital settings. The current study revealed that mothers of children with ASD experienced anxiety and depressive symptoms. In addition to social workers, healthcare institutions need to create mechanisms to provide psychological support for the mothers to reduce related emotional challenges or problems.

## 5. Conclusions

The current study illuminates the experience of mothers raising a child with ASD from the mothers’ perspectives. The diagnosis of their child with ASD was not only emotionally difficult, as several participants moved through a grief-like process in response, but was compounded by the mothers’ lack of information on ASD and their limited access to psychosocial support from healthcare professionals. Through a meaning-making process, several participants searched for the causes of ASD and struggled to define ASD on their own terms. This process led to cognitive reframing and acceptance. Acceptance of their child’s neurodevelopmental condition appears particularly critical in facilitating the mothers’ transition from despair to hope and in contributing to learning new mothering practices and adopting coping strategies. Importantly, a new mothering role articulated by participants, which entailed reframing goals, adjusting behaviors and developing new life skills and lifestyles, changed their worldview and methods of relating to their child. New mothering is both due to positive learning experiences in raising their children and contributes to new learning, including developing crucial skills in raising child with ASD, such as time management, patience, and empathy. Finally, the findings suggest that the recognition of new mothering and mothers’ meaning-making processes, their stress, coping mechanisms and resilience are essential to informing policies, programs, and counseling and other therapeutic efforts.

## Figures and Tables

**Figure 1 ijerph-20-05333-f001:**
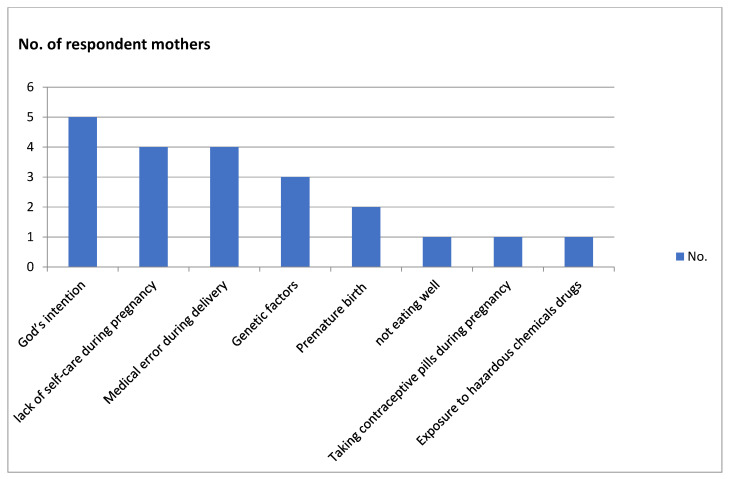
Mother’s causal explanations for ASD. Source: authors’ compiled interview transcripts.

## Data Availability

The data presented in this study are available on request from the corresponding author on a reasonable request. The data are not publically available due to privacy and ethical issues.

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
