# Peer review of "Towards a “New Mothering” Practice? The Life Experiences of Mothers Raising a Child with Autism in Urban Ethiopia"

_ijerph, 2023, doi:10.3390/ijerph20075333_

Round 1
Reviewer 1 Report
Congratulations on tackling such an important topic.
I have some suggestions:
1) lines 109 to 124 need to be cancelled
2) Line 182: "All participants provided oral consent to take part in this study"
Considering that in Table 1 personal data have been published, even if under pseudonyms, why was no written consent requested?
Author Response
Congratulations on tackling such an important topic.
I have some suggestions:
- lines 109 to 124 need to be cancelled
- We apologize for this error which occurred when moving the document into the journal template, it has been corrected.
- Line 182: "All participants provided oral consent to take part in this study". Considering that in Table 1 personal data have been published, even if under pseudonyms, why was no written consent requested?
- We have clarified that verbal consent was obtained, recorded and transcribed as approved by the research ethics board at Addis Ababa University
- Given concerns raised about confidentiality, the table of participants has been removed and a summary of partipants added to replace the more specific information to enhance confidentiality of participants
Reviewer 2 Report
Below are some comments that may strengthen the overall impact of this paper.
1. P.3 line 109-124 all the words should delete that are not relevant to this study.
2. The prevalence of ASD in low or middle income countries should be reviewed and provided the estimates in the introduction section. Below the reference could be considered or find more epidemiologic studies.
---Hossain MM, Khan N, Sultana A, Ma P, McKyer ELJ, Ahmed HU, Purohit N. Prevalence of comorbid psychiatric disorders among people with autism spectrum disorder: An umbrella review of systematic reviews and meta-analyses. Psychiatry Res. 2020 May;287:112922.
3. I recommend that the introduction section should review more studies regarding the issues of family condition and parenting style (focusing on maternal raising) in other rural community, low social environment, low or middle income country who affected by ASD. Furthermore, making more comparisons for the mothering issues between Ethiopia or other countries that may have similar cultural background.
There are some references could be considered:
(1) Blake JM, Rubenstein E, Tsai PC, Rahman H, Rieth SR, Ali H, Lee LC. Lessons learned while developing, adapting and implementing a pilot parent-mediated behavioural intervention for children with autism spectrum disorder in rural Bangladesh. Autism. 2017 Jul;21(5):611-621.
(2) Stewart LA, Lee LC. Screening for autism spectrum disorder in low- and middle-income countries: A systematic review. Autism. 2017 Jul;21(5):527-539.
4. Based on the results of this study, I hope to see more directions toward the aspect of intervention issues in Ethiopia. Authors may draw some future aspects and speculation in discussion section.
Author Response
- 3 line 109-124 all the words should delete that are not relevant to this study.
- Again, we apologize for this error which occurred when moving the document into the journal template, it has been corrected.
- The prevalence of ASD in low or middle income countries should be reviewed and provided the estimates in the introduction section. Below the reference could be considered or find more epidemiologic studies.
Hossain MM, Khan N, Sultana A, Ma P, McKyer ELJ, Ahmed HU, Purohit N. Prevalence of comorbid psychiatric disorders among people with autism spectrum disorder: An umbrella review of systematic reviews and meta-analyses. Psychiatry Res. 2020 May;287:112922. - The introduction section has been significantly re-written to include a discussion of prevalence including the reference kindly suggested by the reviewer
- I recommend that the introduction section should review more studies regarding the issues of family condition and parenting style (focusing on maternal raising) in other rural community, low social environment, low or middle income country who affected by ASD. Furthermore, making more comparisons for the mothering issues between Ethiopia or other countries that may have similar cultural background.
There are some references could be considered:
(1) Blake JM, Rubenstein E, Tsai PC, Rahman H, Rieth SR, Ali H, Lee LC. Lessons learned while developing, adapting and implementing a pilot parent-mediated behavioural intervention for children with autism spectrum disorder in rural Bangladesh. Autism. 2017 Jul;21(5):611-621.
(2) Stewart LA, Lee LC. Screening for autism spectrum disorder in low- and middle-income countries: A systematic review. Autism. 2017 Jul;21(5):527-539.
- Additional information regarding ASD in low and middle income countries has been added to the rewritten introduction including specific information about family issues related to ASD in African countries and the references suggested by the reviewer
- Based on the results of this study, I hope to see more directions toward the aspect of intervention issues in Ethiopia. Authors may draw some future aspects and speculation in discussion section.
- The introduction contains additional information about interventions in Ethiopia which are then revisited in the discussion. As suggested earlier by the editors, we have included an “implications for social work practice” section that is presently 6 paragraphs, we can expand further if required.
Reviewer 3 Report
Although there is a wealth of international data on the impacts of caregiving for individuals with ASD, there is a shortage of data regarding these impacts on caregivers in low resources settings. This is a fascinating study in this respect. Most studies found parents of a child with ASD had decreased parenting efficacy and increased parenting stress. They increased mental and physical health problems compared with parents' children with other developmental disorders in high-income countries. This is very interesting to understand this impact in less affluent non-western societies.
Hence, some issues need to be resolved in the present study. First of all, the referencing style was according to APA, and as far as I am informed, the considered journal does not support this style.
The second undeniable issue is the leftover of journal template, which has negatively impacted the flow of the text. For example, there is a nonrelevant text for lines 109 to 124 where the method and material are supposed to be presented.
As an international reviewer and researcher in this field, I was curious to see one section devoted to the Ethiopian contest and its culture. Some information is distributed in different areas, but allocating one particular section might be more helpful.
I expected to see some research questions at the end of the introduction to serve the findings in the discussion section.
There is a very vague aim for the study (to contribute to the growing research 105 that documents parental experiences and coping mechanisms in raising a child with Autism in the global south, specifically shedding light on the experiences of Ethiopian mothers.) Hence, presenting open-ended research questions is recommended. As some examples …What are mother's thoughts on their child's Autism? How does it feel to care for a child with Autism? How do they think about social attitudes and feelings about them as mothers of a child with Autism or….
I also noticed that the numbing of the titles and subtitles are not following a logical sequence (after presenting the method and materials, which was numbered 2, the presented subtitle was numbered 1.2).
More information regarding the interview protocol is needed, where the mothers meet and the prior arrangement to understand their reaction, any incentives or refreshments and question format and probs and…
I did not find it common to consider a similar title for section 3 (Results). The authors added "New Mothering Experiences" to this section's title. I suggest this is to be presented as a subsection.
At the beginning of the results section (New Mothering Experiences), I suggest general reports regarding the overall extracted themes and subthemes to be presented. Then each theme is explained in more detail.
The information presented in the results section needs to be given before this section (possibly in the introduction section or a separate section explaining the Ethiopian context). As an example, information is presented (lines 253 to 257).
I also did not find the presented graph regarding the mothers' expressed theme very useful in depicting their ideas. I suggest using percentages and frequencies to define maternal impressions clearly. There are some instances in which frequencies have been mentioned, but it is not followed in the entire section. To further analyze the interview content, the frequencies of the answers need to be considered to help gain a better understanding of the findings across maternal interviews. According to Sandelowski (2004), counting and percentage reporting are integral to the analysis process, and numbers in qualitative studies are used to establish the significance of findings. They are also used to recognize patterns and to make analytic generalizations from data.
I also do not understand why the main title of this section (New Mothering Experiences) has been added to theme number 4 (New Mothering: Reframing Relationships with their Child with Autism). If this is only a related theme to item number 4, it is not worth mentioning it as a sub-title, but if the authors believe that it is that significant, they need to justify why it is said only one sub-theme (3.4.)?
In sum, the "results" section needs to be reorganized. It is a combination of presenting the findings and justifications that need to be expressed in the "discussion" part. As an example, in lines 340 and 341 in which, it is said that: "In an emotional and relational sense, a shift in mothering practice is expressed 340 through reframing maternal relations with the child who has Autism. Some participants…" this is a justification and needs to be presented in the discussion—or lines 354 and 355 and also 360 and 370.
Mentioning names such as Meselech and Mastewal said in the table and presented in the result section without any introduction was very odd.
I think more quotations from the mothers in this section will help the reader with the extracted theme. I also think that themes are very general, and there are possibilities for some subthemes to make the findings more applicable.
Make it clear how the adopted theoretical perspective helps you justify the findings.
How did the findings help in answering the study questions or aims?
Autism spectrum disorders (ASD) and Autism are used in this text. I suggest using one to help with the consistency of the used terms.
I also think that the limitation of this study is far beyond what is mentioned presently. The sample size is not this important to be mentioned in this section since 20 members are considered sufficient for a qualitative study (except for a grounded theory-type approach study in which 30 members as an adequate sample is recommended). But I think it is crucial to mention that considering ASD as one diagnosis and underrating its different types is one of the most critical shortcomings of the present study. ASD consists of a very heterogeneous group with very diverse abilities and disabilities. The available findings correlate the caregivers' stress, satisfaction, and general impressions with child functionality, communication levels, and behavioral problems. These are all neglected here.
I suggest a major revision and reorganizing of the study, particularly in the findings and discussion sections.
Author Response
Although there is a wealth of international data on the impacts of caregiving for individuals with ASD, there is a shortage of data regarding these impacts on caregivers in low resources settings. This is a fascinating study in this respect. Most studies found parents of a child with ASD had decreased parenting efficacy and increased parenting stress. They increased mental and physical health problems compared with parents' children with other developmental disorders in high-income countries. This is very interesting to understand this impact in less affluent non-western societies.
- Some issues need to be resolved in the present study. First of all, the referencing style was according to APA, and as far as I am informed, the considered journal does not support this style.
- We apologize for the oversight and have changed the referencing format
- The second undeniable issue is the leftover of journal template, which has negatively impacted the flow of the text. For example, there is a nonrelevant text for lines 109 to 124 where the method and material are supposed to be presented.
- Again, we apologize for this error which occurred when moving the document into the journal template, it has been corrected.
- As an international reviewer and researcher in this field, I was curious to see one section devoted to the Ethiopian contest and its culture. Some information is distributed in different areas, but allocating one particular section might be more helpful.
- Information regarding the context in Ethiopia has been moved to a dedicated section in the introduction which contains additional information
- I expected to see some research questions at the end of the introduction to serve the findings in the discussion section.
- Research questions have been added to the introduction section
- There is a very vague aim for the study (to contribute to the growing research 105 that documents parental experiences and coping mechanisms in raising a child with Autism in the global south, specifically shedding light on the experiences of Ethiopian mothers.) Hence, presenting open-ended research questions is recommended. As some examples …What are mother's thoughts on their child's Autism? How does it feel to care for a child with Autism? How do they think about social attitudes and feelings about them as mothers of a child with Autism or….
- Research questions have been added to the introduction section
- I also noticed that the numbing of the titles and subtitles are not following a logical sequence (after presenting the method and materials, which was numbered 2, the presented subtitle was numbered 1.2).
- The journal template inserted numbers. We agree with the reviewer that this is confusing and have removed the numbering of sub-sections.
- More information regarding the interview protocol is needed, where the mothers meet and the prior arrangement to understand their reaction, any incentives or refreshments and question format and probs and…
- We have expanded the information about the interview format in the methods section.
- I did not find it common to consider a similar title for section 3 (Results). The authors added "New Mothering Experiences" to this section's title. I suggest this is to be presented as a subsection.
- We have altered the titles and numbering of the results section.
- At the beginning of the results section (New Mothering Experiences), I suggest general reports regarding the overall extracted themes and subthemes to be presented. Then each theme is explained in more detail.
- We have added a paragraph at the beginning of the results indicating the overall themes.
- The information presented in the results section needs to be given before this section (possibly in the introduction section or a separate section explaining the Ethiopian context). As an example, information is presented (lines 253 to 257).
- We have moved all information in the results section regarding the Ethiopian context and moved it to the rewritten introduction section.
- I also did not find the presented graph regarding the mothers' expressed theme very useful in depicting their ideas. I suggest using percentages and frequencies to define maternal impressions clearly. There are some instances in which frequencies have been mentioned, but it is not followed in the entire section. To further analyze the interview content, the frequencies of the answers need to be considered to help gain a better understanding of the findings across maternal interviews. According to Sandelowski (2004), counting and percentage reporting are integral to the analysis process, and numbers in qualitative studies are used to establish the significance of findings. They are also used to recognize patterns and to make analytic generalizations from data.
- Numbers of mothers expressing certain views was added as suggested by the reviewer.
- I also do not understand why the main title of this section (New Mothering Experiences) has been added to theme number 4 (New Mothering: Reframing Relationships with their Child with Autism). If this is only a related theme to item number 4, it is not worth mentioning it as a sub-title, but if the authors believe that it is that significant, they need to justify why it is said only one sub-theme (3.4.)?
- We have altered the titles and numbering of the results section.
- In sum, the "results" section needs to be reorganized. It is a combination of presenting the findings and justifications that need to be expressed in the "discussion" part. As an example, in lines 340 and 341 in which, it is said that: "In an emotional and relational sense, a shift in mothering practice is expressed 340 through reframing maternal relations with the child who has Autism. Some participants…" this is a justification and needs to be presented in the discussion—or lines 354 and 355 and also 360 and 370.
- As noted above, we have removed information about Ethiopia from this section and moved it to the introduction, we have also re-named and numbered some of the sections.
- Mentioning names such as Meselech and Mastewal said in the table and presented in the result section without any introduction was very odd.
- We have added the following sentence to the beginning of the results section “Throughout this section pseudonyms are used to put a human face on the experiences of mothers while protecting their identity.” We have also removed the table where names are included.
- I think more quotations from the mothers in this section will help the reader with the extracted theme. I also think that themes are very general, and there are possibilities for some subthemes to make the findings more applicable.
- We have altered the results section as noted above, however, we are unclear as to what other specific suggestions the reviewer may have in this respect.
- Make it clear how the adopted theoretical perspective helps you justify the findings.
- The theoretical frame is “mothering” (Barlow and Chapin). As indicated in the discussion section, our results and analysis build on this concept of mothering to develop the construct of “new mothering”.
- How did the findings help in answering the study questions or aims?
- We believe that this can now be found in the discussion section.
- Autism spectrum disorders (ASD) and Autism are used in this text. I suggest using one to help with the consistency of the used terms.
- For consistency, ASD has been used throughout the text
- I also think that the limitation of this study is far beyond what is mentioned presently. The sample size is not this important to be mentioned in this section since 20 members are considered sufficient for a qualitative study (except for a grounded theory-type approach study in which 30 members as an adequate sample is recommended). But I think it is crucial to mention that considering ASD as one diagnosis and underrating its different types is one of the most critical shortcomings of the present study. ASD consists of a very heterogeneous group with very diverse abilities and disabilities. The available findings correlate the caregivers' stress, satisfaction, and general impressions with child functionality, communication levels, and behavioral problems. These are all neglected here.
The limitations section has been expanded in a manner in which we believe addresses the concern of the reviewer.
Round 2
Reviewer 2 Report
Thanks for author's revision. Looking forward to seeing the publication.
Reviewer 3 Report
This form of the previously submitted manuscript is an improved version. I congratulate the authors who have done their best to address the extensive suggestions and comments I did on the previous draft of the study. I appreciated their endeavor. The study will help to increase international understanding of caregiving for individuals with autism spectrum disorder (ASD) in low resources settings.